# Is Mindfulness-Based Stress Reduction Effective for People with Hypertension? A Systematic Review and Meta-Analysis of 30 Years of Evidence

**DOI:** 10.3390/ijerph18062882

**Published:** 2021-03-11

**Authors:** Ciro Conversano, Graziella Orrù, Andrea Pozza, Mario Miccoli, Rebecca Ciacchini, Laura Marchi, Angelo Gemignani

**Affiliations:** 1Department of Surgical, Medical and Molecular Pathology, Critical and Care Medicine, University of Pisa, 56126 Pisa, Italy; ciro.conversano@unipi.it (C.C.); graziella.orru@unipi.it (O.G.); rebecca.ciacchini@gmail.com (R.C.); laura.marchi83@gmail.com (L.M.); angelo.gemignani@unipi.it (A.G.); 2Department of Clinical and Experimental Medicine, University of Pisa, 56126 Pisa, Italy; 3Department of Medical Sciences, Surgery and Neurosciences, University of Siena, 53100 Siena, Italy; andrea.pozza@unisi.it

**Keywords:** mindfulness, mind–body therapies, blood pressure, hypertension, meditation

## Abstract

*Background*: Hypertension is among the most important risk factors for cardiovascular diseases, which are considered high mortality risk medical conditions. To date, several studies have reported positive effects of mindfulness-based stress reduction (MBSR) interventions on physical and psychological well-being in other medical conditions, but no meta-analysis on MBSR programs for hypertension has been conducted. Objectives: The objective of this study was to determine the effectiveness of MBSR programs for hypertension. *Methods*: A systematic review and meta-analysis of randomized controlled trials examining the effects of MBSR on systolic and diastolic blood pressure (BP), anxiety, depression, and perceived stress in people with hypertension or pre-hypertension was conducted. The PubMed/MEDLINE and PsycINFO databases were searched in November 2020 to identify relevant studies. *Results*: Six studies were included. The comparison of MBSR versus control conditions on diastolic BP was associated with a statistically significant mean effect size favoring MBSR over control conditions (*MD* = −2.029; 95% confidence interval (CI): −3.676 to −0.383, *p* = 0.016, *k* = 6; 22 effect sizes overall), without evidence of heterogeneity (*I*^2^ = 0.000%). The comparison of MBSR versus control conditions on systolic BP was associated with a mean effect size which was statistically significant only at a marginal level (*MD* = −3.894; 95% CI: −7.736–0.053, *p* = 0.047, *k* = 6; 22 effect sizes overall), without evidence of high heterogeneity (*I*^2^ = 20.772%). The higher the proportion of participants on antihypertensive medications was, the larger the effects of MBSR were on systolic BP (*B* = −0.750, *z* = −2.73, *p* = 0.003). *Conclusions*: MBSR seems to be a promising intervention, particularly effective on the reduction of diastolic BP. More well-conducted trials are required.

## 1. Introduction

Cardiovascular disease refers to a class of disorders concerning heart and blood vessels which are considered as the primary cause of death in almost all countries by the World Health Organization (WHO) [1]. Hypertension, a long-term medical condition characterized by persistently elevated blood pressure (BP) in the arterial vessels, is a primary risk factor for cardiovascular disease, and, implicated in over 7.1 million deaths per year, it affects around 35% of the adult population worldwide [2,3]. The prevalence and control of hypertension respectively increases and decreases with advancing age [4]. Epidemiological data show that the prevalence, awareness, and control of hypertension differ across gender: it is well-established that men would have lower levels of hypertension awareness and control and a higher prevalence and incidence of this medical condition as compared with age-matched women before the sixth decade of life [5,6]. 

Current management strategies include drug and/or non-drug interventions [7]. Dependent on risk conditions, different classes of antihypertensive medications are recommended. For prehypertensive individuals, health-promoting lifestyle modifications are indicated [8]. Despite the efficacy of antihypertensive drugs, the prevalence of uncontrolled hypertension in the adult population remains high [9]. The National Institute for Health and Care Excellence (NICE) guidelines [7] emphasize the importance of lifestyle in the management of hypertension and highlight the need for the inclusion of non-pharmacological components in treatment programs, such as healthy diet, regular physical activity, total abstinence from smoking, and limited alcohol consumption. 

Additional behavioral targets, including stress reduction, have been proposed [10]. Stress is a pathological process resulting from psychophysiological responses triggered by threats to homeostasis [11], and chronic psychosocial stress leads to an increase in BP and the risk of developing hypertension [12,13,14]. Moreover, it is now established that chronic conditions frequently present with various types of psychopathology such as depressive aspects, cognitive-behavioral disorders, sleep disorders, and extremely high levels of stress [15,16,17,18,19,20,21,22,23,24,25,26,27]. Psychological corollaries of stress, including increased levels of anxiety, depression, or anger, are known predictors of hypertension [28]. Stress is associated with hyperactivation of the autonomic nervous system, dysregulation of the hypothalamic pituitary adrenal axis, and maladaptive lifestyle factors such as smoking, alcohol use, obesity, and lack of exercise [28,29]. Such evidence encourages the possibility that stress reduction programs could be useful in treating hypertension. In addition, both primary studies and systematic reviews showed that clinically meaningful symptoms of anxiety, depression, and perceived stress are often associated with hypertension, and anxious-depressive disorders represent the psychiatric conditions most common in people with hypertension [30,31,32,33].

BP control should be considered a primary target of both pharmacological and non-pharmacological interventions since the reduction of high-BP is associated with a decreased risk for the development of major cardiovascular diseases [34]. The American Heart Association [34] reviewed the effectiveness of alternative approaches to lowering pressure, including behavioral therapies, such as yoga, transcendental meditation (TM), mindfulness meditation, acupuncture, relaxation techniques, biofeedback, and aerobic exercise. The results showed that meditation, especially TM, may be clinically effective in lowering BP, although the mechanism of action is unclear and detailed recommendations are precluded by lack of evidence [34].

Mindfulness meditation is the core skill of mindfulness-based stress reduction (MBSR), an 8-week, standardized group program, developed by Jon Kabat-Zinn [35]. MBSR is rooted in ancient Buddhist meditative traditions and emphasizes the practice of intentionally focusing awareness on one’s own experience of the present moment in a non-judgmental and non-reactive way. It requires to focus attention to the present-moment, curiosity, openness, and acceptance [36]. The 8-week program usually contains formal practices (body scan meditation, sitting meditation, hatha yoga, and walking meditation) and informal practices (awareness of pleasant/unpleasant daily events, interpersonal communications, repetitive thoughts/emotions, and their connection with bodily sensations) [36].

Initially applied to psychosomatic disorders, MBSR interventions have subsequently demonstrated broad efficacy in improving physical and mental wellbeing in several patient populations [37,38,39,40,41,42,43,44,45,46]. These include beneficial effects on BP. For example, Carlson, Speca, Faris, and Patel [47] observed decreased systolic BP and heart rate in cancer patients who participated in a MBSR program. Similarly, a reduction in systolic BP was also observed in nurses and nursing students who performed 30-min meditation daily for a week [48]. Recent meta-analyses [49,50] highlighted the efficacy of transcendental meditation and yoga in reducing both systolic and diastolic BP in hypertensive adults. Recently, a systematic review [51] identified 5 randomized controlled trials published from 2012 to 2017. The results showed that MBSR groups were associated with greater reductions in systolic and diastolic BP than control groups [51].

## 2. Rationale and Objectives

Overall, there is a shortage of research regarding MBSR interventions in people with hypertension. No study used meta-analytic techniques to summarize the existing evidence. In addition, the available review on this topic summarized the results from studies published in the 2012–2017 timeframe without meta-analytical methods, while the MBSR protocol was published for the first time in 1990 [52]; thus, it might be important to fill in the literature gap and to locate potentially additional studies that might be conducted and published before 2012 and after 2017.

Therefore, the primary objective of the present meta-analysis was to assess the effectiveness of MBSR programs on BP at post-treatment in people with hypertension compared to control conditions (wait-list conditions or treatment as usual, such as education groups about health behaviors, and other interventions based on stress reduction such as relaxation techniques). Since a significant level of heterogeneity was expected across the studies, we explored the potential moderator roles of age, gender, and concurrent antihypertensive drugs, as the literature data previously discussed suggest that hypertension control increases with antihypertensive drugs, it decreases with age, and it is lower amongst men [4,5,6,7,9].

Given the high rates of anxiety, depression, and stress symptoms associated with hypertension [30,31,32,33], as discussed before, a secondary objective of the present review was to summarize the evidence about the effects of MBSR on anxiety, depression, and perceived stress symptoms.

## 3. Method

The present systematic review and meta-analysis was conducted according to the Preferred Reporting Items for Systematic Review and Meta-Analysis guidelines (PRISMA) [53].

### 3.1. Eligibility Criteria

To be eligible for this meta-analysis, studies were required to meet the following conditions, according to the PRISMA guidelines:

(1) Participants: Studies were restricted to adult participants (age ≥ 18 years old) with either elevated BP (120–129/<80 mm Hg), or stage 1 (130–139/80–89 mm Hg) or stage 2 (≥140/90 mm Hg) hypertension. Participants were eligible if they were outpatients or inpatients, they were recruited from any setting, and they were undertaking antihypertensive drugs or they were unmedicated. Participants with comorbid psychiatric and/or medical conditions were not excluded. 

(2) Intervention: Studies were included if they evaluated an MBSR intervention, based on the original program developed by Jon Kabat-Zinn [36], or a mindfulness-based cognitive therapy (MBCT) developed by Segal, Williams, and Teasdale [54], or a mindfulness-based protocol adapted from these and maintaining a comparable format, e.g., eight group sessions of 2–4 h with the practice of mindfulness meditation as the key element.

(3) Comparator: Studies were eligible if they evaluated MBSR compared with either wait-list conditions, treatment as usual, such as education groups about health behaviors, or other interventions based on stress reduction such as relaxation techniques, but not including MBSR techniques/ingredients.

(4) Outcome measures: Studies were included if they assessed systolic and diastolic BP measured with at least two time points (i.e., at pre- and immediate post-test). Follow-up longer than immediate post-test were allowed. Different assessment modalities of BP were considered eligible: clinic and/or ambulatory BP, 24 h BP, day-time BP, and night-time BP. Studies were eligible if they used a variety of measurement devices such as automated oscillometric BP devices or manual sphygmomanometers.

In addition, studies were eligible if they assessed the effects of MBSR on anxiety, depression, and/or perceived stress symptoms. Studies were included if they assessed these outcomes by self-report measures and/or clinician-administered interviews, both with known psychometric properties (i.e., internal consistency values).

(5) Study design: Only randomized controlled trials (RCTs) were eligible.

(6) Language: Studies published in English were included.

(7) Publication timeframe: January 1990–November 2020. This timeframe was chosen because the MBSR protocol was developed and published for the first time in 1990 [16].

(8) Type of article: full-text published in peer-review journals.

### 3.2. Systematic Literature Search

Studies were identified by conducting an online systematic search of the electronic databases PubMed and PsycINFO through the following keywords: “MBSR OR mindfulness OR mind-body therapies AND hypertension OR prehypertension”. We chose these two databases since this search strategy has been considered by some authors [55] to be associated with an optimal combination of databases for literature searches in systematic reviews, as PubMed and PsycINFO are focused on two key domains (medical literature and psychological literature, respectively) related to the research question of the review. The search was carried out by two independent reviewers from 27 to 30 November 2020. In addition, reference lists of the studies included in the meta-analysis were also examined.

### 3.3. Study Selection

Studies were screened against eligibility criteria by two reviewers working independently in two stages. During the first stage, studies were examined with regards to the eligibility criteria after reading the title and the abstract, respectively. During this stage, duplicates were removed. Subsequently, the authors met to compare their selections. Studies were not excluded if there was disagreement between the reviewers on inclusion or exclusion. During the second stage, two authors independently assessed the full text of the papers. At this stage, any disagreements about inclusion or exclusion of studies were discussed and resolved in meetings with another independent reviewer. A last stage consisted of the examination of the reference lists of the studies included in the review to identify studies which had not been located through the search based on the electronic databases.

### 3.4. Data Extraction and Coding

All the information was extracted from each included study and inserted into an Excel worksheet by a reviewer (R.C.). Another independent reviewer (i.e., not involved in the extraction of the data) (A.P.) checked the correctness of the data entered in the worksheet. After entering the data, any discrepancies were discussed in meetings between the two reviewers. The following information was extracted and coded from each study: (1) first author’s surname, (2) publication date, (3) country, (4) inclusion and exclusion criteria, (5) study design, (5) control group, (6) assessment time points, (7) recruitment strategy, (8) treatment, (9) primary outcomes, (10) secondary outcomes, (11) total sample size, (12) diagnosis, (13) percentage of participants on antihypertensive agents, (14) mean age and standard deviation, (15) percentage of females, (16) percentage of participants with medical comorbidities, (17) pre-test and post-test mean BP score and the related standard deviations, and (18) findings related to systolic and diastolic BP, anxiety, depression, and stress between-group comparisons and changes. 

### 3.5. Risk of Bias in Individual Studies

Risk of bias in individual studies was assessed by using the risk of bias assessment tool developed by the Cochrane Collaboration [56]. A reviewer assessed risk of bias independently, then another reviewer evaluated the judgements assigned by the first one, and each discrepancy was discussed and resolved. Each study was judged on six methodological domains considering how certain methodological characteristics might have impacted the results (random sequence generation, appropriate randomization, allocation concealment, treatment allocation, blinding of outcome assessors, blinding of participants and personnel). According to the guidelines provided by Higgins and Green [56], each domain was rated as high, low, or unclear. Risk of bias was classified as low if it was regarded as low by the two independent reviewers for all the domains, as unclear if it was regarded as low or unclear for all the domains, and as high if it was regarded as high for one or more domains.

### 3.6. Meta-Analysis

A random-effect meta-analysis was carried out. The effect sizes for meta-analyses were calculated as differences in means (*MD*) [57], i.e., the mean of all the differences between all sample means, using sample sizes, mean scores, and standard deviations (SD). The use of a *MD* effect-size index allowed a change in BP to be detected through the same unit of measurement across the studies. Effect sizes were calculated by comparing MBSR conditions versus a combined condition which resulted from the combination of the effect sizes derived from both no-intervention (i.e., wait lists) and active control conditions (i.e., progressive muscle relaxation and health education). Negative effect sizes suggested that MBSR was superior as compared with controls. *MD* indices were chosen since BP was assessed through the same measurement unit (i.e., mm Hg) across the included studies. Heterogeneity between the studies was calculated using the *I*^2^ index, which represents a measure of between-study heterogeneity in percentage attributable to variability rather than chance [58]. A value approximating zero indicates homogeneity, whereas values of 0–25%, 25–50%, and 50–100% represent low, acceptable, and high heterogeneity, respectively [58]. The likelihood of publication bias was estimated through the Egger’s test, which is an unweighted regression analysis based on the precision of each study as the independent variable and the effect size divided by its standard error as the dependent variable [59]. A non-statistically significant result of the *t*-test for the null hypothesis of an intercept equal to zero allows us to discard publication bias [59]. In addition, in order to verify the likelihood that the findings are subjected to publication bias, the funnel plot [60] was inspected. 

Multivariate meta-regressions were carried out investigating the effects of gender (coded as the percentage of females), age (coded as the mean age in the study), and anti-hypertensive medications (coded as the percentage of participants on this type of medications).

All the analyses were performed with a 95% confidence interval (CI). The meta-analysis was carried out through the Comprehensive Meta-Analysis software version 3.0 and summarized in the forest plots.

## 4. Results

### 4.1. Study Selection

Initially, the research returned a total number of 32 articles. The total number of articles after duplicates were removed was 30. From this initial pool, we excluded 5 articles, screened based on English language and type of publication, and the remaining 25 full texts were analyzed more closely. After applying eligibility criteria, 19 articles were excluded from the final review. The final number of articles enrolled in this study was 6. Figure 1 presents the PRISMA flow chart depicting the study selection process.

### 4.2. Study Characteristics

An overview of the characteristics of the study samples, selection criteria, outcome measures, and main results related to BP, anxiety, depression, and perceived stress symptoms are shown in Table 1.

### 4.3. Participant and Setting Characteristics

All except one of the studies included patients recruited at private clinics and hospitals [61,62,63,65,66]. One study [69] included adults recruited from a senior housing facility. Five studies [61,62,63,65,66] included both genders, whereas one study [69] included only men. Two studies [61,66] included unmedicated patients with grade-1 hypertension, one study [62] included unmedicated stage-1 hypertension, and one study [63] included cardiac patients (coronary artery disease, mitral valve replacement, other cardiovascular diseases) referred to a specialized cardiac clinic—all the participants received antihypertensive drugs (i.e., angiotensin-converting enzyme inhibitors, angiotensin II receptor blockers, beta-blockers, calcium channel blockers, and diuretics.). One study [69] included older adults aged 62 years or older recruited from a senior housing facility (13 with hypertension, stage I and II, 4 with pre-hypertension, 3 normotensive), and the type of medication was not specified. One study [63] included patients with a diagnosis of coronary heart disease who had been hospitalized or had had symptoms of heart disease within the last 1 year.

### 4.4. Intervention Characteristics

Three studies [61,62,63] applied MBSR, in accordance with the original program of Jon Kabat-Zinn. Two studies [65,66] applied MBCT. One study [69] applied an 8-week mindfulness-based program, designed to teach mindfulness meditation and to develop social and emotional skills. All studies also included daily home practice. All but one study [61,62,63,66,67,68,69] reported information on who led the MBSR group: one psychologist [65], two therapists [62], one therapist [61], one psychiatrist [66], and one instructor [69], all trained in MBSR protocol. One study [65] did not report information about the instructor’s background. All but one study [61,62,63,66,69] reported daily home practice of 15 to 45 min—one study [66] did not specify the length of home meditation practice. Control groups differed widely between studies. Two studies [62,63] used a wait list, one study [61] applied progressive muscle relaxation, one study [69] a social support group, one study [46] a health education session about cardiovascular disease and its management, and one study [66] 8 weekly health education sessions.

### 4.5. Outcome Measures

#### BP Measures

Diastolic and systolic BP were assessed as outcome measures in all studies using two different measurement methods: clinic BP and ambulatory BP. Both methods differed widely from one study to another. All studies assessed clinic BP: three studies used an automated oscillometric BP device [61,62,69], two studies used a manual sphygmomanometer [63,65], and one study [62] used only the ambulatory BP measure. Two studies [61,66] measured both clinic and ambulatory BP, although one study [61] did not report results on ambulatory BP.

Only three studies [63,66,69] reported perceived stress outcome measures and two studies [47,48] reported anxiety and depression outcome measures. Three studies did not include follow-up measures [61,63,69] while three studies [62,65,66] provided 24-week, 3-month, and 4-week follow-up measurements, respectively.

### 4.6. Risk of Bias in Individual Studies

Three studies [62,65,69] were classified as at high-risk of bias, two studies [61,66] were judged as at unclear risk of bias, and one study [63] was considered as at low-risk of bias. An overview of the risk of bias assessments is provided in Table 2.

### 4.7. Meta-Analysis: Effects of MBSR versus Control Conditions on BP

#### 4.7.1. Diastolic BP

The comparison of MBSR versus control conditions on diastolic BP was associated with a statistically significant mean effect size favoring MBSR over control conditions (*MD* = −2.029; 95% CI: −3.676 to −0.383, *p* = 0.016, *k* = 6; 22 effect sizes overall), without evidence of heterogeneity (*I*^2^ = 0.00%). The forest plot is depicted in Figure 2. The funnel plot (Figure 3) and Egger’s test, which did not result statistically significant, suggested absence of publication bias (intercept = 1.060, *t*_(4)_ = 0.979, 1-tailed *p*-value = 0.191, 2-tailed *p*-value = 0.382). 

Subsequently, when the studies associated with a high risk of bias were removed through a sensitivity analysis [57,59,61], the comparison of MBSR versus control conditions on diastolic BP was associated with a statistically significant mean effect size favoring MBSR over control conditions (*MD* = −3.057; 95% CI: −4.030 to −2.084, *p* = 0.000, *k* = 3; 8 effect sizes), without evidence of heterogeneity (*I*^2^ = 0.00%). Egger’s test did not result statistically significant, suggesting absence of publication bias (intercept = −0.305, *t*_(1)_ = 0.494, 1-tailed *p*-value = 0.353, 2-tailed *p*-value = 0.707).

A sensitivity analysis conducted only on the comparisons of MBSR versus wait-list control conditions on diastolic BP showed a non-significant mean effect size (*MD* = −0.944; 95% CI: −4.159 to −2.271, *p* = 0.565, *k* = 2), without evidence of heterogeneity (*I*^2^ = 0.000%). The analyses conducted only on the comparisons of MBSR versus active control conditions (i.e., health education and relaxation groups) on diastolic BP showed a non-significant mean effect size (*MD* = −2.963; 95% CI: −3.954 to −1.973, *p* = 0.108, *k* = 4), without evidence of heterogeneity (*I*^2^ = 0.000%) and publication bias (intercept = 0.748, *t*_(2)_ = 0.991, *p* = 0.213). 

#### 4.7.2. Systolic BP 

The mean effect size for MBSR versus control conditions on systolic BP was statistically significant, although at a marginal level (*MD* = −3.894; 95% CI: −7.736–0.053, *p* = 0.047, *k* = 6; 22 effect sizes overall), without evidence of heterogeneity (*I*^2^ = 20.772%). The forest plot is presented in Figure 4. The funnel plot (Figure 5) and Egger’s test, which did not result statistically significant, suggested absence of publication bias (intercept = −0.636, *t*_(4)_ = 0.376, 1-tailed *p*-value = 0.362, 2-tailed *p*-value = 0.725).

When the studies associated with a high risk of bias were removed through a sensitivity analysis [57,59,61], the comparison of MBSR versus control conditions on systolic BP was associated with a mean effect size favoring MBSR over control conditions, which was not statistically significant (*MD* = −3.544; 95% CI: −9.359–2.271, *p* = 0.232, *k* = 3; 8 effect sizes overall), with evidence of low heterogeneity (*I*^2^ = 38.879%). Egger’s test did not result statistically significant, suggesting absence of publication bias (intercept = −0.448, *t*_(1)_ = 0.100, 1-tailed *p*-value = 0.468, 2-tailed *p*-value = 0.936).

Subsequently, a sensitivity analysis conducted only on the comparisons of MBSR versus wait-list control conditions on systolic BP showed a non-significant mean effect size (*MD* = −5.351; 95% CI: −15.738 to −5.035, *p* = 0.313, *k* = 2), without evidence of heterogeneity (*I*^2^ = 0.000%). The analyses conducted only on the comparisons of MBSR versus active control conditions (i.e., health education and relaxation groups) on diastolic BP showed a non-significant mean effect size (*MD* = −3.218; 95% CI: −8.304 to −1.818, *p* = 0.197, *k* = 4), without evidence of heterogeneity (*I*^2^ = 20.111%) and publication bias (intercept = −0.123, *t*_(2)_ = 0.062, *p* = 0.478). 

### 4.8. Moderator Analyses

Two moderator analyses were carried out separately by multivariate meta-regressions, where mean age, the percentage of females, and the percentage of participants on anti-hypertensive medications were entered as predictors.

Mean age (*B* = 0.175, *z* = 0.94, *p* = 0.174), the percentage of females (*B* = −0.013, *z* = −0.22, *p* = 0.411), and the percentage of participants on anti-hypertensive medications (*B* = −0.015, *z* = −0.74, *p* = 0.230) were not significantly associated with the effect sizes on diastolic BP. 

The percentage of participants on anti-hypertensive medications was negatively and significantly associated with the effect sizes of systolic BP (*B* = −0.750, *z* = −2.73, *p* = 0.003), while mean age (*B* = 0.253, *z* = 0.97, *p* = 0.165) and the percentage of females (*B* = 0.043, *z* = 0.50, *p* = 0.309) were not significantly associated with the effect sizes. This result suggested that the higher the proportion of participants taking antihypertensive medications was, the larger the effects of MBSR programs were. 

An overview of the analyses carried out in the meta-analysis is presented in Table 3.

## 5. Discussion

Hypertension is among the most important risk factors for cardiovascular diseases, a primary cause of death worldwide. The NICE guidelines [7] highlight an important role of stress in the development of this medical problem. MBSR represents an effective strategy for the management of a variety of psychological and medical conditions. A quantitative summary of the available evidence about its effectiveness for people with hypertension has not been conducted yet. 

The present study is the first work using meta-analytical techniques to pool the evidence about the effectiveness of MBSR programs for hypertension. We identified 6 studies, one more study than the most recent review [52], on this topic. The present findings suggest that MBSR is a promising strategy for people with hypertension. Specifically, we found that the effects of MBSR programs were associated with significant mean effect sizes on both diastolic and systolic BP, without evidence of high heterogeneity and publication bias. MBSR resulted more effective than different control conditions, including wait lists, health education, and relaxation groups. This result appears interesting as both systolic and diastolic BP have been considered independent predictors of adverse cardiovascular outcomes [71,72]. Overall, the effects of MBSR on BP are consistent with the data from studies that found a reduction in BP following participation in MBSR programs in other clinical populations, such as patients with cancer, patients with hypertension taking part in transcendental meditation and yoga programs, and in non-clinical populations such as healthcare workers [44,45,46,47,48].

Moderator analyses were not able to identify significant predictors of the effects on diastolic BP, since gender, age, and concurrent antihypertensive medications were not related to the effect sizes on this outcome. However, for systolic BP, we found that the higher the proportion of participants on antihypertensive medications was, the larger the effects of MBSR were, suggesting that the benefits of MBSR might be optimal in combination with antihypertensive drugs. However, since the studies did not randomly assign participants to antihypertensive drugs or antihypertensive drugs + MBSR conditions through an additive design, this conclusion is preliminary and needs further empirical support by future studies aimed to evaluate the superiority of the combination of antihypertensive drugs and MBSR over antihypertensive drugs alone. The lack of moderator effects of age and gender suggests that the extent to which MBSR programs are effective in the management of BP might not differ across age cohorts and gender. Specifically, the fact that age was not a moderator of the effects of MBSR is in contrast with evidence from a previous systematic review [51], where meditation played a noticeable role in decreasing the BP of subjects older than 60 years of age. The lack of moderator effects of age and gender seems to be particularly relevant for clinical practice since older people and men tend to have lower hypertension awareness and control than younger ones and women, respectively [4,5,6]; thus, it seems to be in contrast with the clinical expectation that older people and men would have a less positive response to MBSR on BP.

In the present review, evidence from two studies suggested that MBSR programs were more effective than control conditions on reducing anxiety and depression symptoms, as suggested by self-report measures. In addition, data from three studies showed that MBSR programs were associated with significantly higher changes in perceived stress symptoms. These results are in line with previous evidence from systematic reviews which showed that MBSR interventions are capable to reduce anxious and depressive symptoms and stress levels in chronic diseases such as heart failure [73,74]. Unfortunately, due to the small number of studies, we were not able to pool the data on anxiety, depression, and perceived stress through meta-analyses. 

In conclusion, the present meta-analytic study suggests that MBSR might be a useful strategy aimed to increase the individual’s capacity to manage both diastolic and systolic BP in people with different stages of hypertension. Therefore, MBSR programs might be a valid adjuvant in both the treatment and prevention of hypertension. This appears particularly relevant in times of the COVID-19 pandemic, where most of the constraints on the population relate to physical activity and lifestyle changes [75]. 

## 6. Limitations and Future Directions

The small number of studies is a strong shortcoming of this review and the present results should be considered preliminary. The evidence related to a statistical significance for systolic BP with a *p*-value at a marginal significance level (*p* = 0.047) might be due to insufficient power, which did not allow us to detect a significant effect. Another limitation concerns the fact that only three studies assessed the effects of MBSR at follow-up and this prevented the evaluation of the effects long-term. In addition, we pooled data from studies using different control conditions. Sensitivity analyses that distinguished the comparisons based upon wait lists from those based upon active control conditions (i.e., health education or relaxation groups) did not detect significant mean effect sizes because they were probably underpowered. 

Moreover, it should be considered that two out of the six studies were judged at high risk of methodological bias. This issue was handled through a sensitivity analysis that excluded such studies and found a significant mean effect size for diastolic but not for systolic BP. Overall, these limitations suggest the need for further research based upon larger and good-quality trials using long-term follow-up assessments. 

Another aspect that should be investigated is the role of moderators. The present moderator analyses were not able to identify significant moderators of the effect sizes on diastolic BP. It would be interesting to explore further variables, for example the potential role of comorbid psychiatric disorders. Unfortunately, none of the included studies reported these data to allow us to perform this analysis. In addition, it would be useful to investigate whether a greater reduction in BP associated with MBSR protocols can predict a lower risk of cardiovascular failure or stroke in the long term.

More research is needed about the effects of MBSR on psychological outcomes including anxiety, depression, and perceived stress, since in our review, only three studies assessed perceived stress and two assessed anxiety and depression symptoms, respectively. It would be interesting to examine whether MBSR can have some effects on psychological variables which generally have a key role in the management and prognosis of hypertension, such as quality of life, health-related locus of control, and self-management skills [76,77,78].

Another relevant aspect to be investigated in the future is whether MBSR can be combined with other psychological interventions aimed to improve health status in hypertension, such as lifestyle interventions that promote an optimal level of physical activity [79]. It is also not clear which mindfulness facets were responsible for the reduction in BP levels or also in anxious-depressive and stress symptoms. It might be that the non-judgement facet plays a role in this since it has already been found to be the strongest predictor of anxious-depressive symptoms in cross-sectional studies amongst all mindfulness facets [80,81]. However, in our search, we did not identify process-based studies; therefore, the psychological processes through which MBSR might work for people with hypertension remain still unexplored and future dismantling trials [82] are required to investigate this key point. 

## 7. Conclusions

This is the first meta-analysis which summarized the available evidence about the effectiveness of MBSR programs on BP for people at different stages of hypertension. In six studies, MBSR produced higher changes than control conditions on both diastolic and systolic BP, even if changes on systolic BP were only at a marginal significance level. In studies with a higher proportion of participants on antihypertensive medications, the effects of MBSR on systolic BP were larger. In conclusion, MBSR programs seem to be a promising intervention for people with hypertension, particularly effective on the reduction of diastolic BP.

## Figures and Tables

**Figure 1 ijerph-18-02882-f001:**
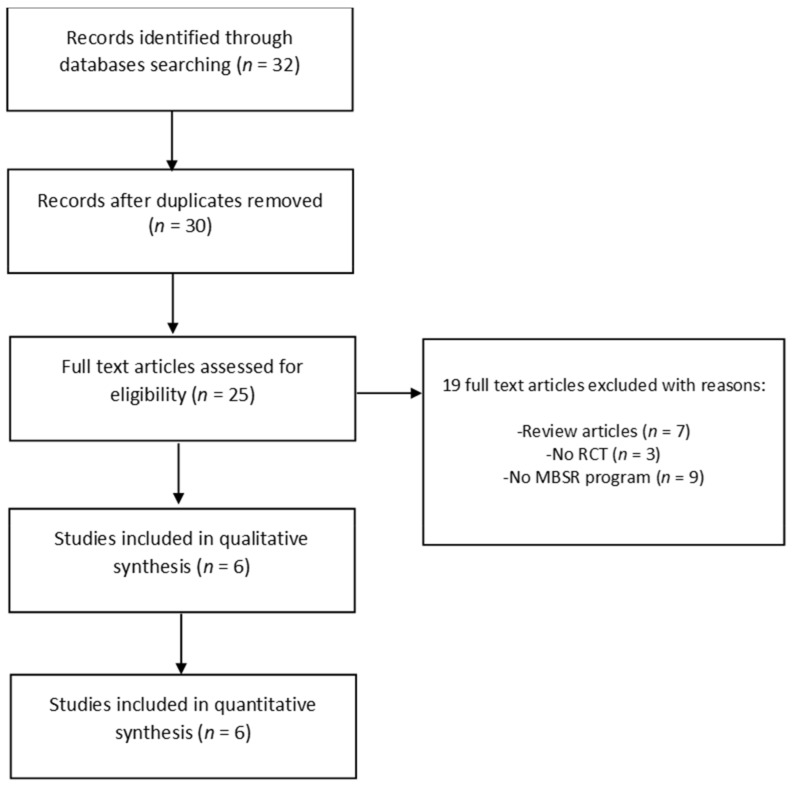
Preferred Reporting Items for Systematic Review and Meta-Analysis (PRISMA) flowchart of the study selection process. Abbreviations: MBSR = mindfulness-based stress reduction, RCT = randomized controlled trial.

**Figure 2 ijerph-18-02882-f002:**
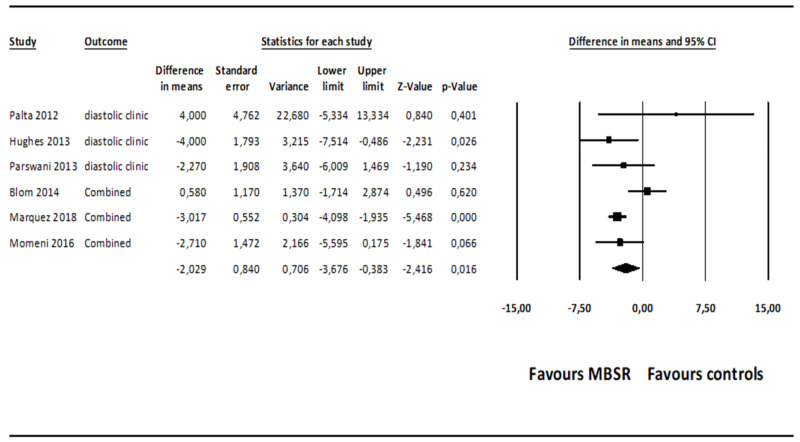
Forest plot of the effect sizes of MBSR versus control conditions on diastolic BP. Abbreviations. BP = blood pressure, MBSR = mindfulness-based stress reduction.

**Figure 3 ijerph-18-02882-f003:**
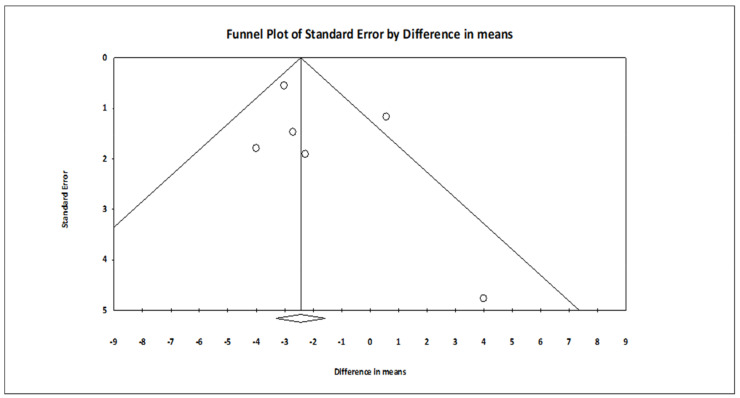
Funnel plot of the effect sizes of MBSR versus control conditions on diastolic BP.

**Figure 4 ijerph-18-02882-f004:**
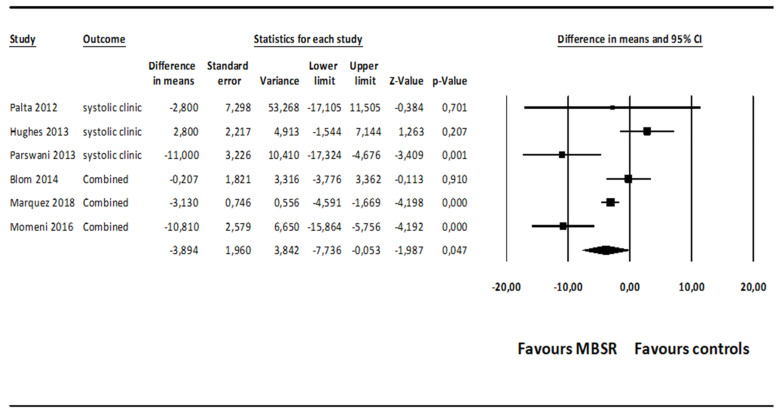
Forest plot of the effect sizes of MBSR versus control conditions on systolic BP.

**Figure 5 ijerph-18-02882-f005:**
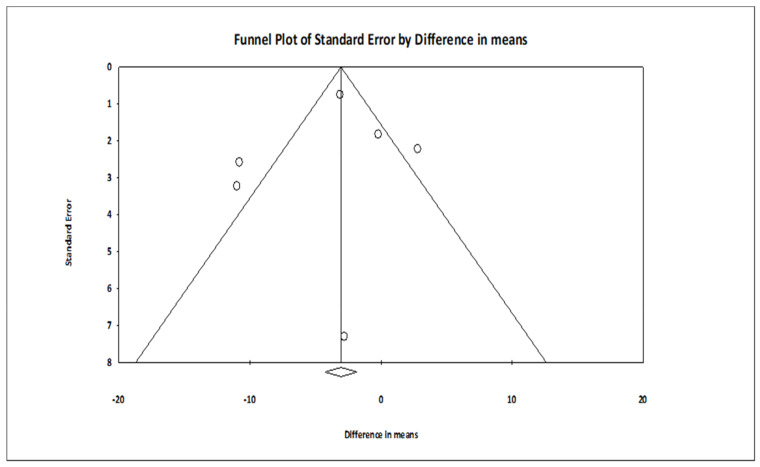
Funnel plot of the effect sizes of MBSR versus control conditions on systolic BP.

**Table 1 ijerph-18-02882-t001:** Characteristics of the included studies (*n* = 6).

First Author	Publication Date	Country	Inclusion (IC) and Exclusion (EC) Criteria	Study Design	Control Group	BP Assessment Time Points	Recruitment Strategy	Treatment	BP Outcomes	Anxiety, Depression, Stress Outcomes	Total Sample Size (Experimental Group; Control Group)	Diagnosis	PercentAge of Participants on Antihypertensive Agents	Mean Age (SD)	Percentage of Females	Percentage of Participants with Medical Comorbidities	Main Findings
Hughes [61]	2013	USA (OH)	IC: age between 30 and 60 years old. EC: being pre-hypertensive, taking antihypertensive medication, being experienced with meditation practices, being current smokers, having any disease (e.g., myocardial infarction, heart failure, chronic kidney disease, diabetes).	RCT	Active control condition including 8 group sessions of progressive muscle relaxation training	(1) PRE-TEST: Initial BP screening administered in a three-week period. (2) PRE-TEST: Clinic and ambulatory BP. (3) POST-TEST: Clinic and ambulatory BP.	Advertisement Participants were enrolled in 7 cohorts ranging in size from 3–11.	MBSR (8 weekly group sessions, each 2.5 h long)	Clinic SBP/DBP (seated BP reading automated oscillometric BP device). Ambulatory SBP and DBP: 24 h monitoring, ambulatory BP device.	Not reported	56 (28; 28)	Elevated BP in the prehypertensive range (SBP 120–139 mm Hg or DBP 80–89 mm Hg)	0	50.3 (6.5)	Total: 57% IG: 61% CL: 54%	Not specified	MBSR: statistically significant reductions in the primary outcomes of clinic SBP and DBP. MBSR is more effective in lowering elevated BP than an active control (PMR).
Blom [62]	2013	CANADA (ONT)	IC: aged 20 to 75 years with mean awake ambulatory systolic or diastolic BP ≥ 135/85 mm Hg or mean 24 h ambulatory BP ≥ 130/80 mm Hg (stage 1 hypertension). BP was required to be <160/100 mm Hg on both office and ambulatory measurements. Participants were naive to antihypertensive medication for at least 6 months before the baseline screening visit.	RCT	Wait list	(1) PRE-TEST (24 h ambulatory BP monitoring); (2) POST-TEST: 12 week after baseline (24 h ambulatory BP monitoring). (3) STUDY CLOSE: all participants returned 24 weeks after baseline for a third 24 h ambulatory BP recording.	Participants were recruited from referring physicians, advertisements in local newspapers, and posters at local hospitals. Study subjects were not given incentives for participation in the study but were reimbursed for parking.	MBSR (8 weekly group sessions, each 2.5 h long)	Clinic SBP/DBP (change in awake and 24 h ambulatory BP using an automated office BP measurement device, BpTRU).	Not reported	101 (50; 51)	Stage 1 hypertension (mean S/DBP ≥ 135/85 mm Hg or mean 24 h ambulatory BP ≥ 130/80 mm Hg)	0	IG: 57 (12); CL: 55 (11)	IG: 64% CL: 63%	Not specified	MBSR did not significantly lower ambulatory BP when compared with BP change in a wait-list control group.
Momeni [63]	2016	IRAN (KASC)	IC: any cardiac diagnosis, suffering from hypertension, receiving antihypertensive agents, not having experienced new heart attacks or cardiac symptoms in the last six months before the study, age ranging from 35 to 60 years, able to participate in the study, high school diploma or higher degrees. EC: suffering from renal problems, diabetes mellitus, or active malignant conditions such as cancer, having a history of convulsion or epilepsy during the last six months prior to the study, being a pregnant or breastfeeding woman, having a history of drug abuse, having received psychological therapies during the month before the study, and having the history of using yoga, meditation, or Zen exercises.	Single-blinded RCT	Wait list	(1) PRE-TEST: DBP and SBP. (2) POST-TEST (same measures).	The study participants were recruited from all cardiac patients referring to a specialized private cardiac clinic located in Kashan, Iran, from April to June 2015.	MBSR (8 weekly group sessions, each 2.5 h long)	Clinic SBP and DBP: auscultatory method and AOBP method	PSS-14 [64]	60 (30; 30)	Diagnosed cardiac problem	100	47 (7); IG: 49.16 (6.31); CL: 46.16 (6.27)	42% (IG: 43.3%, CL: 40%)	Coronary artery disease: 65%; Mitral valve replacement: 21%; Other CVD: 13%	Significant difference between the study groups regarding the post-test values of systolic BP, perceived stress. However, the study groups did not differ significantly in terms of diastolic BP
Palta [65]	2012	USA (MD)	IC: African Americans aged 62 years or older who were living in the building at the time of the baseline interview and had no plans to move. English language specking required. Consent form and baseline questionnaires filled.	RCT	Social support group Two research assistants guided the group by recording attendance and introducing the topic to be discussed by members of the group, but no mindfulness teaching or practices were offered during these meetings. To initiate communication among participants, the research assistants provided pre-planned conversation starters that mirrored the topics in the intervention group. Following the session, participants were offered a fruit and vegetable snack identical to the foods served to the intervention group.	(1) PRE-TEST: demographic information, PSS (Cohen, 1983) and blood pressure measurements with electronic blood pressure machine (2) POST-TEST: blood pressure measurements (same instrument)	Participants were recruited from a low-income senior housing facility in Baltimore City, through informational sessions, flyers, and tabling. This facility had not previously been exposed to any mindfulness programs. All participants were compensated for their time with a $25 gift card upon completion of each survey time point.	ELDERSHINE, mindfulness-based group (8 weekly group session, each 90 min long)	Clinic SBP and DBP	Not reported	20 (12; 8)	not requested	90	IG: 72.3 (4.4) CL: 73.7 (5.8)	95%	Not specified	Comparing the differences between post-intervention and baseline measurements, individuals in the intervention group exhibited a 16.70 mmHg lower diastolic blood pressure compared to the social support control group and this value was statistically significant (*p* = 0.003).
Màrquez [66]	2018	SPAIN	IC: hypertension, EC: medical history of symptomatic heart failure, or left ventricular ejection fraction symptomatic heart failure, cerebrovascular disease or any other condition that might result in death before study completion; patients concomitantly using BP-modifying drugs (cyclosporine, nonsteroidal anti-inflammatory drugs (NSAIDs), steroids, vasoconstrictors, etc.); pregnant women; patients participating in another clinical trial, and patients with previous experience of mindfulness, meditation, yoga, tai chi, chi kung, or similar techniques.	RCT	Active control condition Weekly health education over the same period.	(1) baseline (pre-intervention) visit (2) mid-point visit at week 4 (3) post-intervention visit, at week 8 (4) follow-up visit at week 20	Recruited from among hospital employees and patients from hypertension unit with high-normal BP or grade 1 hypertension	8-week MBCT	Clinic and ambulatory SBP, DBP	PSS-10 [64], POMS [67], DASS-21 [68]	42	High-normal BP or grade 1 hypertension	69.9%	56.5 (7.77)	52.17%	Not specified	At week 8, the intervention group had statistically significant lower ABPM scores than the control group, health education groups, (124/77 mmHg vs 126/80 mmHg (*p* < 0.05) and 108/65 mmHg vs 114/69 mmHg (*p* < 0.05) for 24 h and night-time systolic BP (SBP), respectively) and also had lower clinically measured SBP values (130 mmHg vs 133 mmHg; p = 0.02). At week 20 (follow-up), means were lower in the intervention group (although not statistically significant). Significant post-intervention differences were observed between the two groups, specifically in terms of lower intervention group levels of anxiety (*p* = 0.02) and stress (*p* = 0.05), as measured by DASS-21, and of depression (*p* = 0.02), fatigue (*p* = 0.03) and confusion (*p* = 0.02), as measured by the POMS subscales.
Parswani [69]	2013	INDIA (KARN)	IC: age between 30 and 65 years old, hospitalized or had symptoms of heart disease within the last 1 year and echocardiography test showing ejection fraction >35% with ability to read, write and speak English language. EC: a clinical history suggestive of psychoses, obsessive compulsive disorder, mental retardation, mania, severe depression, neurological or serious medical conditions, and those with previous exposure or currently receiving any psychological intervention.	RCT	Control Group Patients were given health education about coronary heart disease and its management	(1) BASELINE (2) POST INTERVENTION (3) FOLLOW UP	Patients were recruited from the inpatient and outpatient services of St. Johns Medical College and Hospital, Bangalore	MBSR (8 to 10 weekly group sessions, each 1.5 h long)	Clinic SBP and DBP (sphygmomanometer)	HADS [70], PSS [64]	30	Coronary heart disease (CHD)	100	IG: 47.27 (12.15), CL: 50.60 (8.21)	0%	Not specified	Systolic BP was significantly lower in the MBSR group than in the TAU group at post-assessment. No significant difference in DBP. The reduction in DBP was maintained at follow-up and patients showed further decrease. The scores of MBSR group on anxiety, depression and overall distress measured by the HADS were significantly lower than the TAU group. Perceived stress reduced significantly within patients of the MBSR group, and there was a significant difference between the two groups on this dimension at the end of the intervention.

Notes: Abbreviations: PMR: progressive muscle relaxation training; RCT: randomized control trial; BP: blood pressure; SBP: systolic blood pressure; DBP: diastolic blood pressure; BMI: body mass index; CHD: cardiovascular disease; AOBP: automated oscillometric blood pressure, PSS-10 = Perceived Stress Scale, PSS-14 = Perceived Stress Scale, DASS-21 = Depression and Anxiety Scale-21, POMS = Profile OF Mood States, HADS = Hamilton Depression Rating Scale, MBSR = Mindfulness-Based Stress Reduction, RCT = Randomized Controlled Trial.

**Table 2 ijerph-18-02882-t002:** Quality assessment of included studies.

	Selection Bias Domains		Performance Bias	Detection Bias	Attrition Bias	Reporting Bias
Study	Random Sequence Generation	Allocation Concealment	Blinding of Participants and Personnel	Blinding of Outcome Measures	Incomplete Outcome Bias	Selective Reporting
Hughes et al. [61]	?	?	?	?	+	+
Blom et al. [62]	+	−	+	−	+	+
Momeni et al. [63]	+	+	+	+	+	+
Palta et al. [65]	?	−	−	?	+	+
Màrquez et al. [66]	?	?	?	?	−	−
Parswani et al. [69]	+	?	?	?	+	+

Notes: “+” = low risk of bias, “-”= high risk of bias, “?” = unclear risk of bias.

**Table 3 ijerph-18-02882-t003:** Overview of the analyses.

Type of Analysis	Outcome	*k*	*MD*	*p*-Value	95% CI	*I^2^*	Evidence of Publication Bias
**Random-effect meta-analyses**							
Pooled effect sizes	Diastolic BP	6	−2.029	0.016	−3.676 to 0.383	0.000	No (intercept = 1.060, *t*_(4)_ = 0.979, *p* = 0.191)
Pooled effect sizes	Systolic BP	6	−3.894	0.047	−7.736 to 0.053	20.772	No (intercept = −0.636, *t*_(4)_ = 0.376, *p* = 0.362)
**Sensitivity analyses**							
Excluding the studies with high risk of bias	Diastolic BP	3	−3.057	0.000	−4.030 to −2.084	0.000	No (intercept = −0.305, *t_(_*_1)_ = 0.494, *p* = 0.353)
Excluding the studies with high risk of bias	Systolic BP	3	−3.544	0.232	−9.359 to 2.271	38.879	No (intercept = −0.448, *t*_(1)_ = 0.100, *p* = 0.468)
**Sensitivity analyses**							
MBSR versus wait list	Diastolic BP	2	−0.944	0.565	−4.159–2.271	0.000	N/A
MBSR versus wait list	Systolic BP	2	−5.351	0.313	−15.738–5.035	0.000	N/A
MBSR versus active control conditions	Diastolic BP	4	−2.963	0.108	−3.954–1.973	0.000	No (intercept = 0.748, *t*_(2)_ = 0.991, *p* = 0.213)
MBSR versus active control conditions	Systolic BP	4	−3.218	0.197	−8.304–1.818	20.111	No (intercept = −0.123, *t*_(2)_ = 0.062, *p* = 0.478)
**Moderators (multivariate meta-regression)**	**Outcome**		***B***	***z*** ** (** ***p*** **-value)**			
Mean age	Diastolic BP	6	0.175	0.94 (0.174)			
Percentage of females			−0.013	−0.22 (0.411)			
Percentage of participants on anti-hypertensive medications			−0.015	−0.74 (.230)			
**Moderators (multivariate meta-regression)**	**Outcome**						
Mean age	Systolic BP	6	0.253	0.97 (0.165)			
Percentage of females			0.043	0.50 (0.309)			
Percentage of participants on anti-hypertensive medications			−0.750	−2.73 (0.003)			

Notes: BP = blood pressure, *k* = number of studies included in the analysis, MBSR = mindfulness-based stress reduction, MD = mean difference, CI = confidence interval, N/A = not applicable due to the small number of studies.

## Data Availability

Not applicable.

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
