# Peer review of "Is Mindfulness-Based Stress Reduction Effective for People with Hypertension? A Systematic Review and Meta-Analysis of 30 Years of Evidence"

_ijerph, 2021, doi:10.3390/ijerph18062882_

Round 1

Reviewer 1 Report

Please check if the abstract is structured or not for this journal. 

As well as the type of citation used

Authors contributions and acknowledgmnts are missing. 

Over all, the mauscript seems well conducted. After some minor spelling and typing mistakes, i would encourage authors to profund on the discussion part. 

Reviewer 2 Report

The choice of topic is very interesting especially in a covid era where most of the constraints in the population relate to physical activity and lifestyle changes. There are already many researches that have investigated the effectiveness of physical activity on mental health in adulthood and by practicing, for example, even a simple activity such as dance, as demonstrated by Vaccaro et al., 2019. Therefore, I believe that proposing a study that emphasizes the importance of practicing regular physical activity is fundamental, even more so if it is a mindfullness technique whose benefits are in this paper marked.

However, there are limitations in the present study.

First I think it is necessary to describe in more detail the selection of researches and later to be clearer in the description of the meta-analysis section.
Figure 1 appears clear but for a more immediate reading you are asked to repeat the description of the abbreviations shown in a note or caption. 

Finally, it would have been appropriate to specify the type of dressing in section 4.3

Reviewer 3 Report

I applaud the authors on a well conducted systematic review and meta-analysis as well as a very well written manuscript. This work contributes significantly to a growing body of work on the effects of mindfulness. As noted by the authors, there is significant heterogeneity and issues of methodological rigor in prior studies. Thus, a review of this quality serves to inform management of hypertension as well as provides significant and specific guidance for future research. 

Reviewer 4 Report

The aim of this study was to research meditation, the central skill of stress reduction interventions based on mindfulness in people with hypertension. The authors emphasized that no study used meta-analytical techniques to summarize the existing evidence. However, the authors used only two electronic databases PubMed and PsycINFO. Authors must justify why they have stopped using other electronic databases for this topic.
